# The Influence of Different Extraction Methods on the Structure, Rheological, Thermal and Functional Properties of Soluble Dietary Fiber from Sanchi (*Panax notoginseng*) Flower

**DOI:** 10.3390/foods11141995

**Published:** 2022-07-06

**Authors:** Guihun Jiang, Karna Ramachandraiah, Zhaogen Wu, Kashif Ameer

**Affiliations:** 1School of Public Health, Jilin Medical University, Jilin 132013, China; wzg12345.ok@163.com; 2School of Lifesciences, Sejong University, Seoul 05006, Korea; karna@sejong.ac.kr; 3Institute of Food Science and Nutrition, University of Sargodha, Sargodha 40100, Pakistan

**Keywords:** Sanchi flower, soluble dietary fiber, extraction methods, functional properties

## Abstract

The influence of different extraction methods, such as acidic (AC), enzymatic (EN), homogenization (H), ultrsonication (U) and alkali (AL), on structure, rheological, thermal and functional properties of soluble dietary fiber (SDF) from Sanchi flower was evaluated in this study. The highest extraction yield (23.14%) was obtained for AL-SDF extract. Glucose (Glc) and galactose (Gal) were found to be the major constituents in Sanchi SDF. Homogenization and Ultrsonication treatments caused significant compaction of pores in the microstructures. FTIR analysis showed increased hydrolysis of pectin and hemicellulose in U, AL and AC-SDF extracts. H-SDF and AC-SDF exhibited similar shear rate change with the rise in shear stress. H-SDF was thermally more stable than other SDF extracts. Among all extraction methods, H-SDF and U-SDF exhibited the highest water holding capacity (WHC), oil-holding capacity (OHC), Bile acid-adsorption capacity (BAC), Cholesterol-adsorption capacity (CAC) and Glucose adsorption capacity (GAC). Thus, Sanchi flower SDF with improved functional properties could be utilized as a functional food ingredient in the development of various food products.

## 1. Introduction

Sanchi/Sanqi (*Panax notoginseng*), also known as Chinese ginseng, belonging to the genus *Panax*, is a dicotyledonous perennial plant [1]. The term ginseng is commonly used in reference to the species *Panax ginseng* Meyer, and particularly to the root of the species. The genus panax encompasses 12 species that include *Panax notoginseng* [2]. The root of P. notoginseng is known for its therapeutic effects, which are employed in the treatment of cerebral vascular disease and coronary heart disease. While the beneficial effects of P. notoginseng root have been studied, other parts of the plant, such as stem, leaf, sideslip and flower have also shown therapeutic effects. In particular, water extracts of Sanchi flower had shown the strongest effects among the different parts. The exerted effects included the capacity to protect rat pheochromatocytoma (PC12) cells from the cytotoxicity of H_2_O_2_ and inhibit the deleterious effects of xanthine oxidase [3]. In another study, flowers of P. notoginseng were shown to provide pharmacological benefits such as hepatoprotective effect [4]. The medicinal properties of *P. notoginseng* have been attributed to flavonoids, saponins and polysaccharides [3]. 

Dietary fibers (DF) are non-starch polysaccharides that contain a minimum of 10 monomeric units, which cannot be hydrolyzed by the enzymes (endogenous) present in the intestines [5]. DF have been classified into water insoluble DF: IDF (e.g., lignin, cellulose, hemicelluloses) and water soluble DF: SDF (e.g., pectin, β-glucans, fructooligosaccharides). However, studies have shown that SDF provide higher health benefits than IDF. Hence, several approaches have focused on the increased extraction of SDF from various plant materials. Nonetheless, DF have been reported to improve physiological functions such as lipid profile, blood cholesterol and insulin sensitivity. DF also contribute to the prevention of major diseases such as colon cancer, diabetes and obesity. In particular, soluble dietary fibers (SDF) have attracted attention due to its solubility, gelling potential, oil/water holding capacity and surface properties [6]. Hence, European Food Safety Authority (EFSA) and Food and Drug Administration (FDA) have recommended a daily intake of about 25–38 g for an adult [7]. Consequently, SDF extracted from various plant materials have been utilized as additives/enhancers that improve technological quality (e.g., gelling agent, stabilizer and encapsulating agent). However, it has been shown that the technological properties of SDF are contingent upon molecular weight, molecular weight distribution, structure and concentration [6]. 

Common extraction methods that are currently used to extract SDF mainly include physical, chemical and enzymatic extractions [8]. Nonetheless, in physical methods, ultrasonication (U) has several merits, including low energy and water consumption, lower maintenance costs and reproducibility. Ultrasonication uses cavitation forces to generate shock waves that cause destruction of cell walls [9]. A simple method that also uses less energy is homogenization (H) wherein high velocity, impact and shear forces cause the breakage of cell walls [10]. In chemical methods, acid extraction and alkali extraction are widely used despite drawbacks such as solvent residues and environmental pollution. The wide usage of acid or alkali-based extraction is attributed to merits such as simple operation and reduced production costs [11]. While enzymatic extraction is also a simple and low energy requiring method, the high cost is a major disadvantage. Furthermore, costs have been shown to increase with an increasing number of enzymes. Nonetheless, enzymatic extraction is utilized to break and release constituents that are bound in the interior of plant materials, which are less accessible by commonly used methods [12]. 

The selection of extraction method has been suggested to be based on simplicity, efficiency (yield, processing time), adaptability (to different materials), reproducibility, safety, cost-effectiveness and sustainability [9,13]. Moreover, different extraction method types are known to alter the SDF components and structural characteristics, thereby influencing its physicochemical and functional characteristics [11]. Nonetheless, to the best of our knowledge, there are no published reports available regarding the effects of different extraction methods on structural characteristics and functional properties of Sanchi flower SDF. Therefore, this study was aimed at the evaluation of different extraction methods, namely, ultrasonication (U), enzymatic (EN), homogenization (H), alkaline (AL) and acidic (AC) on the extractability of SDF from Sanchi flower. Furthermore, the effects of various extraction methods on microstructural, rheological, thermal, physicochemical and functional characteristics were also evaluated.

## 2. Materials and Methods

### 2.1. Materials

Sanchi flower was purchased from a local supermarket in Jilin, China. All chemicals and reagents used in this study were of analytical grade.

### 2.2. Preliminary Treatment of Sanchi Flower

The Sanchi flower samples were washed and dried overnight in an oven (40 °C). After that, the dried samples were ground using a disc-mill (FZ102, Taisite Instrument Co., Ltd., Tianjin, China) and sieved (60 mesh) to obtain Sanchi flower powders, which were dried at room temperature and stored in a desiccator.

### 2.3. SDF Extraction from Sanchi Flower

#### 2.3.1. Acid (AC) Extraction 

A method described by Yuliarti et al. [14] was followed (with some modifications) to extract SDF from Sanchi flower, using 1% (*w*/*v*) citric acid. In this type of extraction, Sanchi flower powder was added and mixed with citric acid (ratio 1:40 (*w*/*v*)) on a water bath (40 °C) for 2 h. Centrifugation (3000× *g*) of the resultant acidic solution was performed for 15 min using a centrifuge. Following centrifugation and supernatant collection, 95% ethanol (four-fold volumes) was added and incubated for 2 h under ambient temperature to collect the residues. The residues were then washed with 95% ethanol, followed by oven drying at 60 °C for 5 h to obtain the AC-SDF. The extraction yield (%) of AC-SDF was calculated using the following formula:Y (%) = C/W × 100%(1)where C denotes AC-SDF weight and W stands for Sanchi flower powder weight.

#### 2.3.2. Alkali (AL) Extraction

SDF were extracted based on a method described by Zhang et al. [15] after certain modification, utilizing 5% (*w*/*v*) NaOH solution. Four hundred milliliters of NaOH solution was added to 10.0 g Sanchi flower powder, and the resultant solution was placed in a water bath at 40 °C for 2 h, followed by centrifugation (3000× *g*) for 15 min using a centrifuge. Later, the supernatants were collected, 95% ethanol was added, and they were incubated for 2 h. One hundred percent ethanol was used to wash the resulting residues, which were oven dried at 60 °C for 5 h to get AL-SDF. The extraction yield (%) of AL-SDF was calculated using the aforementioned Formula (1).

#### 2.3.3. Enzymatic (EN) Extraction

Enzymatic extraction of SDF from Sanchi flower was undertaken according to a method described by Kurek et al. [16]. Ten grams of Sanchi flower powder was placed in a beaker and one hundred milliliters of distilled water at pH 9.5 (pH adjusted using 0.1 mol NaOH) was added. The samples were mixed on a rotator for 1 h with shaking. Then, using a centrifuge, the samples were subjected to centrifugation (3000× *g*) for 10 min and the supernatant was collected. To degrade starch, the pH value of the collected supernatant was adjusted to the optimal value, pH = 7.0 (using 0.1 mol HCl), and the thermostable α-amylase (Thermamyl SC, Novozymes, Denmark) was added. Later, the solution was incubated with enzymes at 80 °C. After cooling the solution down, the pH was lowered to 3.5 using 2 M acetic acid with the aim of reducing protein solubility. Isoelectric precipitation of proteins was performed at the pH range 3.5–4.0 followed by heating, which was undertaken using a water bath. Subsequently, a centrifuge was used to perform centrifugation at 5000× *g* for 20 min at 4 °C. Later, the supernatants were collected, and EN-SDF were obtained as described in the previous section. The extraction yield (%) of EN-SDF was then determined according to Formula (1).

#### 2.3.4. Homogenization (H) Assisted Extraction

To extract Sanchi flower SDF by homogenization, samples were dispersed in water (10:400, *w*/*v*) first, and then the suspension was heated at 80 °C in a water bath for 30 min. The suspension was homogenized using an Ultra-Turrax homogenizer (T25 Basic, Digital, Ika, Staufen Germany) for 25 min at the speed of 7000 RPM at 25 °C. The mixture was centrifuged at 5000 g for 15 min, and the residue was then collected and dried to obtain H-SDF. The extraction yield (%) of H-SDF was then calculated using Formula (1).

#### 2.3.5. Ultrasonication (U) Extraction 

To extract Sanchi flower SDF via ultrasound, 10 g of Sanchi flower powder was added in a beaker with 400 mL of distilled water. Then, the solution mixture was placed in an ultrasonic water bath (RK510H, 35 kHz, Sonorex, Bandelin, Germany) at 50 °C for 30 min followed by centrifugation at 5000× *g* for 15 min. Later, the supernatants were collected and oven-dried at 60 °C for 5 h to obtain U-SDF. The extraction yield (%) of U-SDF was then determined according to Formula (1).

### 2.4. Protein Content

The protein content of Sanchi flower SDFs samples were analyzed as per the recommended AOAC [17] methods using *N* × 6.25 as a conversion factor.

### 2.5. Color Analysis

Sanchi flower SDFs samples were analyzed for their color properties (L*, a* and b* values) with help of a colorimeter (CM-3600 A, Konica Minolta, Osaka, Japan). CIELAB color space was used with +a* indicating redness and −a* indicating greenness. While +b* denotes yellowness and −b* blueness, L* refers to darkness or lightness (L = 100; white and L = 0; black).

### 2.6. Environmental Scanning Electron Microscopy (ESEM)

Using an ESEM (Quanta 250 FEG, FEI Company, Hillsboro, OR, USA), the morphology and microstructural features of Sanchi flower SDFs were determined. On the horizontal plane, the samples were sliced into rectangular shape. The samples were placed on a specimen holder and sputter coated with gold. The microstructural examination was performed at an operational voltage of 5 kV. The images were captured at 500× and 1000× magnification. 

### 2.7. Fourier-Transform Infrared (FT-IR) Spectroscopy

The analyses of organic functional groups of Sanchi flower SDFs samples were undertaken using a FT-IR spectrophotometer (Tensor 27, Bruker Daltonics Inc., Bremen, Germany). The spectrum wavelength was 400–4000 cm^−1^ at 4 cm^−1^ resolution with 4 scans at a scan speed of <10 s. Spectra were obtained in Attenuated Total Reflection (ATR) mode.

### 2.8. Thermal Properties 

The thermogravimetric analyzer (TGA 550, TA Instruments, New Castle, DE, USA) was employed to evaluate SDFs (5 mg) samples according to a method described by Du et al. [18] under the certain conditions. The temperature range of 30–600 °C and heating rate of 20 °C/min under nitrogen atmosphere were utilized. 

### 2.9. Rheological Properties

#### 2.9.1. Solution Preparation

The SDFs extracted via different methods were dissolved in a 20 mM sodium phosphate buffer (pH 6.5) to prepare 1% solutions mixture (*w*/*v* on dry weight basis) using a magnetic stirrer for 10 min at ambient temperature. While heating the solutions to 80 °C, the solutions were constantly stirred and the temperature maintained for 15 min. After heating, the solutions were cooled down to an ambient temperature with constant stirring. Finally, the solutions were stored at 4 °C for 24 h before rheological properties were evaluated. 

#### 2.9.2. Rheological Measurements

The rheological properties of SDFs solutions were evaluated utilizing a rheometer (Discovery HR-1, TA, New Castle, DE, USA). The analysis was performed using steel cone geometry (diameter = 60 mm and gap = 56 μm). Assessment of rheological properties of samples was performed using continuous shear test with shear rate ranging between 0.10 to 1000 s^−1^ at 25 °C. With increasing shear rate from 0.10 to 1000 s^−1^, viscosities of samples were measured. At a temperature of 25 °C, measurements of all samples were undertaken three times. The apparent viscosity at 2001/s (η), flow behavior index (n) and consistency coefficient (K) of Sanchi flower samples were measured. Experimental data were fit to the Power Law Model as per the following equation:η = Kγ^n−1^(2)where η denotes apparent viscosity (Pa·s^−1^), K denotes consistency coefficient (Pa·s^n^) and n is the flow behavior index (dimensionless).

### 2.10. Monosaccharide Compositions

The 1-phenyl-3-methyl-5-pyrazolone (PMP) was used to label released monosaccharides. Each sample (2 mg) was dissolved in 1 mL of 2 mol/L trifluoroacetic acid (TFA) and hydrolyzed under a sealed condition at 120 °C for 2 h in a hydrothermal reactor. Later, the residue was washed by co-evaporation using ethanol to remove the excessive TFA. The hydrolysate solution was then mixed with 200 μL methanol solution (0.5 mol/L) of PMP (methanol as solvent) and 200 μL of NaOH solution (0.3 mol/L) and then incubated for 60 min at 70 °C. Two hundred microliters of HCl (0.3 mol/L) was used to neutralize, thereby quenching the reaction. The product was then extracted (thrice) with chloroform (1 mL). Before HPLC analysis, a 0.45 μm membrane filter was used to filter the aqueous layer containing PMP-labeled derivative. The Dionex Thermo Ultimate 3000 HPLC system (Dionex Co., Sunnyvale, CA, USA) fitted with an Ultimate 3000 diode array detector (DAD, Thermo Fisher Sci., Waltham, MA, USA) was used to detect PMP labeled monosaccharides. A mixture of acetonitrile (A) and 0.1 mol/mL phosphate buffer solution (PBS, pH 6.7) in a ratio of 82:18 (*v*/*v*) was used the mobile phase. Analyses were performed on a Supersil ODS2 column (5 μm, 4.6 × 250 mm^2^) with an injection volume of 20 μL with 0.8 mL/min of flow rate at 30 °C. At a detection wavelength of 245 nm, different types of monosaccharides were used as reference standards (glucose, mannose, glucuronic acid, galacturonic acid, galactose, rhamnose, xylose, arabinose, ribose and fucose).

### 2.11. Hydration Properties

The water holding capacity (WHC) and oil holding capacity (OHC) were determined according to methods described by Meng et al. [5]. Briefly, a certain amount (0.5 g) of sample was placed in a centrifuge tube (10 mL). Five milliliters of distilled water was mixed with the samples, which were then incubated for 1 h at 37 °C. Afterwards, water was filtered out and the WHC values of the samples were measured using the following equation:WHC (g/g) = (W2 − W1)/W1(3)where W1 stands for the total dry weight of the sample and W2 for the total weight of the sample after water has been filtered out. 

The determination of OHC was carried out by mixing the sample (0.5 g) with soybean oil in a centrifuge tube that was later incubated for 1 h at 37 °C. Following the centrifugation of the sample at 5000× *g* for 10 min, excess soybean oil was discarded by inverting the centrifuge tube. OHC was then quantified as follows: OHC (g/g) = (W2 − W1)/W1(4)where W1 denotes the total dry weight of the sample and W2 denotes the total weight of the sample after the excess oil has been removed.

### 2.12. Bile Acid-Adsorption Capacity (BAC)

BAC was measured based on a method described by Luo et al. [19]. One gram of SDF sample was mixed with thirty microliters of sodium cholate (1–3 mg/mL, pH 7.0) and incubated at 37 °C for 2 h. Absorbance of the supernatant was measured using a spectrophotometer (UV-1800, Shimadzu Instruments Mfg. Co., Ltd., Kyoto, Japan) at 620 nm. BAC was expressed as mg/g.

### 2.13. Cholesterol-Adsorption Capacity (CAC) 

CAC was determined according to on a method described by Luo et al. [19]. About 1 g of each SDF samples was mixed with 30 mL of diluted yolk solution (using 9 volume of distilled water, a fresh egg yolk was diluted and homogenized). In a single egg yolk, the total amount of cholesterol contained about 275 mg [20]. After adjusting the pH to 2.0 or 7.0, the samples were incubated for 180 min at 37 °C. Absorbance of the supernatant was measured at 550 nm. CAC was expressed as mg/g.

### 2.14. Glucose Adsorption Capacity (GAC) 

SDFs samples (0.5 g) were mixed with 50 mL glucose solution (at a concentration of 50 mmol/L). After continuously stirring for 2 h (at 37 °C), centrifugation of the samples was performed for 5 min at 2862× *g*. The final glucose concentration of the collected supernatant was determined using a glucose assay kit (oxidaseperoxidase kit, Rongsheng Biotech Co., Ltd., Shanghai, China). GAC was expressed as mg/g.

### 2.15. Statistical Analysis 

All experiments were undertaken in triplicates. One-way analysis of variance (ANOVA) was used to determine variances between treatments by SPSS version 18.0 (Chicago, IL, USA). The differences between the means were evaluated using the Duncan’s multiple-range tests for means with 95% confidence limit (*p* < 0.05).

## 3. Results

### 3.1. SDF Yield, Protein Content and Color Values

The extraction yields, protein contents and color values of SDF extracted via different methods are provided in Table 1. The highest extraction yield (23.14%) was for SDF extracted by alkali (NaOH). The highest yield (*p* ≤ 0.05) for AL-SDF could be due to the thorough destruction of the cell walls by NaOH, which could have also partly dissolved the hemicellulose present in IDF, resulting in its conversion to SDF, thereby increasing the total yield [21]. However, the other extraction methods resulted in comparable yields (*p* > 0.05) for AC-SDF, U-SDF, H-SDF and EN-SDF. The highest protein content was obtained for EN-SDF (2.77 %) and U-SDF (2.45%) compared to other SDFs. Among all extraction methods, H, AC and AL treatments resulted in lower protein content owing to the degradation of the cellulose, lignin and hemicellulose that caused facilitated protein removal. It is also likely that these methods could have caused protein degradation, which led to reduced protein contents [22]. In our previous study, Sanchi flower was also reported to contain about 550 mg/100 g and 115 mg/100 g of total phenolic content (TPC) and total flavonoid content (TFC), respectively. While saponin content was about 17 g/100 g, minerals such as calcium (Ca), sodium (Na), zinc (Zn), iron (Fe) and manganese (Mn) were estimated to be about 65 μg/g, 60 μg/g, 35 μg/g, 11 μg/g and 10 μg/g, respectively [23]. For the color values, the highest L* value was observed in the case of U-SDF whereas samples obtained by EN extraction exhibited lower L* values than other extraction methods. With respect to food industry, the higher degree of lightness is reported to be more feasible in the preparation of various products, such as sauces, dips and soups. Furthermore, the highest a* (redness) and b* (yellowness) values were obtained for EN-SDF and AL-SDF. This could be attributed to the presence of high amounts of impurities, such as denatured proteins and starches [6].

### 3.2. Microstructure

The microstructures of Sanchi flower SDF as analyzed by means of ESEM are demonstrated in Figure 1. It was evident from ESEM micrographs that AC-SDF and EN-SDF exhibited (Figure 1A,C) more compacted structures with lesser degree of pore loosening as compared to AL-SDF (Figure 1B), H-SDF (Figure 1D) and U-SDF (Figure 1E). Furthermore, the cracks and fissures were less visible in AC-SDF and EN-SDF; however, H, U and AL treatments significantly (*p* < 0.05) increased the irregularity owing to severe impairment and loosening of SDF microstructures. It has already been reported by Wang et al. [8] that DFs with loose spatial structure exhibit rises in specific surface area, which might exert influence on adsorption capacities of oil, water, glucose, nitrite ion and bile acid. Therefore, it could be inferred on the basis of ESEM results that the structures of Sanchi flower SDF undergo alteration depending on the extraction method.

### 3.3. FT-IR Spectroscopy

The FT-IR spectra of SDF extracted by different extraction methods are demonstrated in Figure 2A. In all the SDF extracts, the broad peaks in spectral regions of 3480 and 3414 cm^−1^ corresponded to the stretching vibrations of hydroxyl (OH^−^) groups of hemicellulose and cellulose. The decreased intensity in this spectral range could be due to the breakage of intermolecular bonds [24]. The stretching vibrations in the spectral region of 2927 cm^−1^ indicated the presence of CH groups embedded in the sugar methyl and methylene groups. The weak absorption spectral bands at IR regions of 2923 and 2856 cm^−1^ corresponded to the stretching vibrations of C–H groups, which are typically observed in the case of polysaccharides polymers [24]. The spectral peak arising in the spectral regions ranging from 1200 to 1400 cm^−1^ was attributable to the variable angle vibrations of CH. In addition, the absorption peaks in this aforementioned range are considered as the characteristic FT-IR absorption peaks of SDF-derived saccharides. The FT-IR spectral peaks in the vibrational regions ranging from 1000 to 1300 cm^−1^ corresponded to the contraction vibrations of C–O vibrations, whereas spectral absorption peaks in the range from 700 to 1000 cm^−1^ corresponded to the presence of characteristic peaks of α and β pyran monosaccharides [16]. FTIR results confirmed that homogenization led to less peak broadening as compared to U-SDF, EN-SDF, AL-SDF and AC-SDF. The peak morphology was significantly different in H-SDF and EN-SDF as compared to AC-SDF, AL-SDF and U-SDF, implying that U, AL and AC treatments led to increased hydrolysis of pectin and hemicellulose. In all the SDF extracts, the IR characteristic peak at 1000 cm^−1^ indicated C–O stretching vibration in C–O–C linkages (typical IR absorption peak of xylan), which indicated that SDF is comprised of xylan hemicellulose. IR peak at spectral region of 1660 cm^−1^ was observed in H-SDF, U-SDF and AC-SDF, which corresponded to stretching or bending vibration of aromatic lignin hydrocarbon, whereas this characteristic peak was absent in the case of AL-SDF and EN-SDF, implying that AL and EN treatments caused enhanced degradation of lignin molecules. The visibility of peak around the spectral region of 870 cm^−1^ suggested the presence of β-glycosidic linkages in SDF extracted samples, a characteristic of polysaccharides, which is in agreement with a previous study on amaranth [16]. 

### 3.4. Thermal Properties

Figure 2B illustrates the TGA analyses of Sanchi flower-derived SDF. The TGA curves of AC-SDF, AL-SDF, EN-SDF, H-SDF and U-SDF were classified into three stages. At the initial temperature range of 30 to 210 °C, all SDF samples exhibited declining tendency in terms of weights with devolatilization occurring at 120 °C [15]. While weight loss was more prominent (*p* < 0.05) in the case of AC-SDF, AL-SDF showed the least degree of weight loss, which occurred in a gradual manner with corresponding rises in temperature. This initial weight loss was ascribed to the possible evaporation of absorbed water from SDF extracts. The recorded mass rate loss for AC-SDF, AL-SDF, EN-SDF, H-SDF and U-SDF were 11.06%, 5.39%, 9.29%, 9.41% and 8.51%, respectively. It was evident from the results of TGA that maximum degree of weight loss occurred during the second stage at temperature range from 210 to 400 °C. This could be due to the elevated pyrolytic polysaccharide (mainly hemicelluloses and pectic polysaccharides) degradation [15]. Moreover, SDF samples extracted by U and AC treatments showed high degree of weight loss, followed by EN-SDF, whereas the weight loss during third stage in case of U-SDF was relatively slower and gradual, implying that H-SDF was thermally more stable as compared to AC-SDF, U-SDF and EN-SDF. This suggested that AC, EN and U extraction methods could lead to unstable thermal properties at elevated temperature. These results were in agreement with the findings of Wang et al. [8], who reported significant influence of extraction methods on thermal stability of kiwifruit (Actinidia deliciosa) DF.

### 3.5. Rheological Properties

The values of consistent coefficient, flow behavior index and apparent viscosity of SDF extracted by different extraction methods are presented in Table 2. Among all extraction methods, EN-SDF exhibited the highest (*p* < 0.05) consistency coefficient (K). The flow behavior index (n) of a fluid is indicative of the degree of non-Newtonian characteristics. If flow behavior index is greater than 1, then it depicts the dilatancy of the SDF extracts. Dilatant, also known as shear-thickening fluids, are known to show higher apparent viscosity with increasing shear rate. In this regard, EN-SDF exhibited the lowest dilatancy [6]. AC-SDF and H-SDF exhibited higher (*p* < 0.05) flow behavior indices of 3.724 and 3.498, respectively. In the case of apparent viscosity (η), AC-SDF exhibited the highest value of 0.317, whereas AL and EN had the lowest values, 0.131 and 0.091, respectively. It is important to note that hydrogen bonds and charge-transfer complexes (CTC) that are formed between polymer chains influence apparent viscosity [25]. However, as shown in Figure 2C, the shear rate changes were determined as a function of applied shear stress. The EN-SDF exhibited rapid changes in the shear rate at an initial shear force level of 28 Pa. AL-SDF showed modifications in shear rate at an applied shear stress level of 55 Pa. On the other hand, when shear stress was applied in the range of 10–88 Pa, U-SDF exhibited slow gradual changes in the shear rate. Conversely, H-SDF and AC-SDF exhibited similar rate of change in shear rate as function of corresponding increases in shear stress from 10 to 120 Pa. This could be due to the major structural changes in SDF fibers extracted by H and AL treatments. In other studies, structural modifications have been attributed to the formation of sheet-like multi-branched structures via the assembly of spherical particles [26]. Furthermore, it has been shown that an extraction method such as homogenization lowers particle size (surface area increment), allowing greater interaction between particles and thereby enhancing the apparent viscosity [26]. As also observed in other studies, H and U treatment could have caused alterations in the cluster structure of SDF, which increased the amount of soluble component contents in the dispersed SDF fraction, thereby increasing the apparent viscosity [26,27]. Additionally, exposure to H treatment could have destroyed the SDF molecular structure by breaking the partial branching patterns as well as by reducing the tightness of internal molecular force and linkages [26].

### 3.6. Monosaccharide Composition

The results of monosaccharides compositions of Sanchi flower SDF are given in Table 3. It was evident from the chromatograms (Figure 3) that regardless of the extraction type, molar ratios of Glc and Gal were found to be higher (*p* < 0.05) in SDF than those of Xyl and Ara. The higher molar ratio of Glc indicated that DF may comprise of starch, hemicelluloses and cellulose [8]. Studies have shown that glucose originates from cellulose and starch. However, in this study, H-SDF exhibited the highest molar ratio of Glc. The higher Glc contents in H-SDF could be ascribed to the promotion of hydrolysis of cell wall cellulose by the H treatment. The decreased Glc contents of U-SDF and AC-SDF implied that the exposure to sonication treatment and acidic environments did not have comparable hydrolytic effects on cell wall celluloses. Moreover, the lack of presence of rhamnose and galacturonic acid indicated that pectin is not a major component of Sanchi SDF [8]. Nonetheless, H-SDF also exhibited the highest molar ratio of Gal. AL-SDF did not show any presence of Gal, which implied that exposure to AL environment resulted in the complete degradation of Gal. Similarly, the lower amounts of Gal in U-SDF and AC-SDF could also be due to significant degradation of galacto-oligosaccharides. Nonetheless, the relatively higher Ara content in H-SDF indicated that most of the cellulose was degraded possibly because of glycosidic bonds breakage, which consequently led to the release of hemicellulose to some extent [8].

### 3.7. Hydration and Functional Properties

#### 3.7.1. WHC and OHC 

The major hydration properties of SDF include WHC and OHC. The WHC represents the water retention ability of food materials after subjection to various processing techniques, such as compression or centrifugation. However, OHC of natural DF is indicative of the degree of maintenance of oil after mixing DF with oil followed by centrifugation [8]. In this study, the effects of different extraction methods on hydration properties of Sanchi flower SDF are presented in Table 4. H-SDF exhibited the highest (*p* < 0.05) WHC and OHC among all samples. In another study, WHC was found to be higher for SDF extracted from papaya peel by ultrasonication than those by alkali treatment [15]. Moreover, chemical treatments (AC and AL) resulted in lower WHC and OHC values. Apart from particle size, surface areas and densities [8,28], WHC is reported to be influenced by the hydrophilic sites available in SDF [8,29]. On the other hand, OHC is dependent on several factors, such as overall electrical charge density, surface properties and hydrophobicity [8]. In this study, SDF extracted by H exhibited the highest WHC and OHC, which could be due to the changes in structures, particle sizes, availability of hydrophilic sites and electrical charge densities [8]. As shown in SEM image (Figure 1D), the H-SDF had more loosened structure and grooves than SDF derived by other treatments. Higher WHC and OHC have also been observed in papaya peel SDF with looser structures [15]. Furthermore, shifts in absorption peaks in FTIR spectra have been shown to be associated with higher WHC [30]. Although, decreased wavelength (FTIR) intensities were observed in AC-SDF and AL-SDF along with H-SDF, lower WHC could be due to comparatively lesser presence of grooves, pores and loosened structures in AC-SDF and AL-SDF. However, another factor that has been shown to influence OHC is protein content. Proteins are reported to possess stronger lipophilic characteristics (in comparison with polysaccharides), higher protein content is linked with increased OHC [6]. In this study, higher OHC of H-SDF and U-SDF could be due to higher protein content. Comparatively lower OHC with EN-SDF could be related to the lesser presence of loosened structures or porosity. Hydration has been shown to be influence many factors, including availability of hydrophilic sites and electrical charge densities [8]. Contrarily, in a study on the oil binding capacity of pineapple pomace, enzyme-extracted SDF had higher OHC than SDF extracted by other methods, such as acidic, homogenization and ultra-sonication [31]. Nevertheless, the high WHC and OHC of SDF implied that Sanchi flower can be a good source of DF for manufacturing food products. Since SDF with improved hydration properties show reduction of water syneresis, it can be utilized in formulated food products and in enhancing the emulsifying properties in high-fat foodstuffs [8,29].

#### 3.7.2. BAC

Studies have shown that the capacity of DF to bind to bile acids (BAs) has implications on the incidence of cardiovascular disease (CVD) [8]. Increased binding of DF to BAs augments its elimination as well as cholesterol–bile acid conversion, thereby lowering the blood cholesterol level [8]. However, in this study, H-SDF and U-SDF exhibited the higher (*p* < 0.05) BAC values of 3.610 and 3.618%, followed by AC-SDF (3.593%), EN-SDF (3.565%) and AL-SDF (3.231%) (Table 4). It was implied from the results that SDF extracts obtained by different methods exhibited different BAC. In another study, the binding capacities of pineapple pomace SDF extracted by homogenization and acid treatment were higher than those of SDF extracted by ultrasonication and enzyme treatment [31]. Decrement in the intensity of absorbance peaks (FTIR) has been linked to alterations in surface properties [32]. Hence, the lowest BAC values for AL-SDF could be due to the changes in surface properties. However, it has also been reported in other investigations that DF gel properties and anionic group content may exert significant effect on the BAC. Moreover, BAC could be correlated to surface properties, particle sizes and internal SDF structures. [8].

#### 3.7.3. CAC

The CAC values of SDF extracts derived from different methods are presented in Table 4. CAC estimates the total mass of cholesterol absorbed by the SDFs, which can undergo physical or chemical adsorption. In physical adsorption, the particle size, porosity, surface area and temperature play an important role. In chemical adsorption, electrostatic charges and hydrophobicity are considered important [33]. However, it has been reported in earlier studies that absorption of cholesterol by DF can lead to a reduction in serum cholesterol levels. In this study, at pH 2 (simulating stomach pH), H-SDF and AL-SDF exhibited the highest CAC value of 12.533 and 12.725%, respectively, followed by U-SDF (11.99%), EN-SDF (11.86%) and AC-SDF (11.61%). At pH 7 (simulating small intestine pH), AL-SDF exhibited the highest CAC value of 12.82%, whereas H-SDF had the lowest 10.43% (Table 4). In another study, homogenization and ultrasonication resulted in pineapple pomace SDF with higher CAC than those treated with acid and enzymes [31]. Nonetheless, the influence of multiple factors on CAC has been observed in other studies on bamboo shoot DF [19] and pea DF [34]. Lower CAC values at low pH could be due to the repulsion between positive charges of DF and H+ [34]. Lower CAC values at pH 7 could be related to decreased polar groups and increased space obstacles [35]. Nevertheless, Sanchi flower SDF possess the capacity to adsorb cholesterol in both the small intestine and stomach. Binding capacities of SDF to oil, sodium cholate and cholesterol indicate the adsorption capacity of SDF for lipophilic constituents [31]. 

#### 3.7.4. GAC

GAC is a parameter that is used to evaluate the DF absorption capacity of glucose. In the intestines, DF possess the capacity to bind to glucose, in turn reducing the postprandial serum glucose. Hence, GAC is considered an important functional property of SDF [8]. Furthermore, fibers have been shown to adsorb glucose in the gastrointestinal system. Increased GAC values suggest that the SDF has greater capacity to absorb glucose during the entire GI transit time. It has been shown in other studies that increased surface areas and number of cavities could have contributed to the increased GAC values [5]. Nonetheless, among all of the extracts, U-SDF exhibited the highest GAC value (96.62%), followed by AL-SDF (83.37%), H-SDF (75.15%), AC-SDF (73.52%) and EN-SDF (51.29%) (Table 4). However, extraction methods such as homogenization, ultrasonication, acidic and enzymatic treatment in pineapple pomace SDF with comparable GAC [31]. Nevertheless, in this study, the higher GAC values observed with U treatment could be attributed to the increased viscosity, porosity and specific surface area of SDF, which in turn increased the ability of SDF to trap glucose molecules. It may also be implied from these results that both DF type and the type of extraction method may exert significant effect (*p* < 0.05) on the GAC of SDF derived from the Sanchi flower.

## 4. Conclusions

In this study, five SDF extraction methods—acid, enzyme, homogenization, ultrasonication and alkaline—were employed for the extraction of SDF, which were characterized in terms of structure, rheological, physicochemical and functional characteristics. It was evident from the results that AL-SDF had the highest extraction yield among all samples. SEM micrographs showed that AC-SDF, EN-SDF and AL-SDF exhibited more compacted structures with lesser degree of pore loosening when compared to H-SDF and U-SDF. H-SDF and AC-SDF exhibited similar shear rate change with rise in shear stress. H-SDF was thermally more stable as compared to other SDF extracts. Homogenization treatment caused greater enhancement of the hydration properties (WHC and OHC) than other treatments. Glucose adsorption capacity (GAC) and Bile acid-adsorption capacity (BAC) were also impacted by the different treatments with ultrasonication treatment resulting in elevated values. Although H-SDF had relatively lower CAC potential in small intestines (than stomach) and lower GAC, homogenization (H) can be considered an effective extraction method. Therefore, processing Sanchi flower via H and U methods could result in SDF with optimal functional properties, which could be utilized by the food industry. While selection of extraction method is contingent upon many factors, these treatments could add value to the production of food formulations derived from Sanchi DF. Future studies have to focus on the influence of extraction methods on antioxidant activities of Sanchi flower. Another area worth studying is the long-term effect of extracted Sanchi SDF on human health.

## Figures and Tables

**Figure 1 foods-11-01995-f001:**
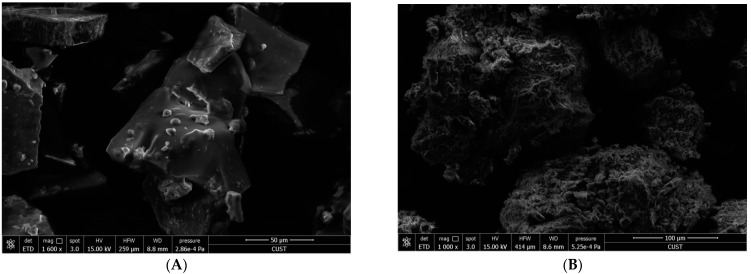
SEM images for AC-SDF (**A**), AL-SDF (**B**), EN-SDF (**C**), H-SDF (**D**) and U-SDF (**E**). AC, Acid; Al, Alkali; EN, Enzyme; H, Homogenization; U, Ultrasonication.

**Figure 2 foods-11-01995-f002:**
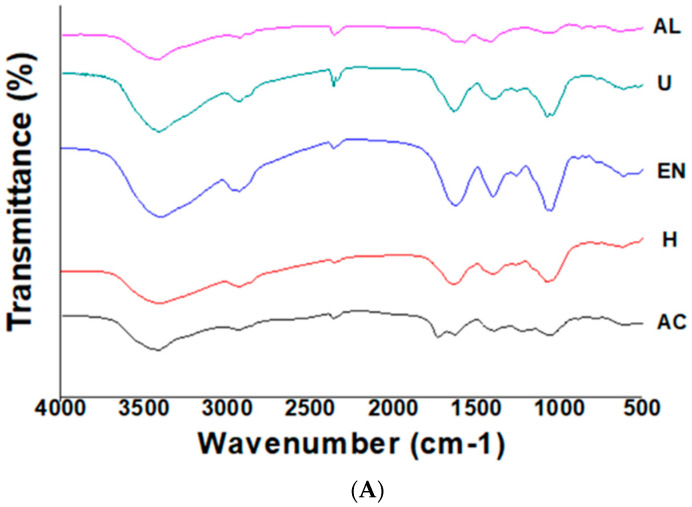
FT−IR spectra (**A**) thermal properties (**B**) and rheogram plot (**C**) for Sanchi flower. AC, Acid; AL, Alkali; EN, Enzyme; H, Homogenization; U, Ultrasonication.

**Figure 3 foods-11-01995-f003:**
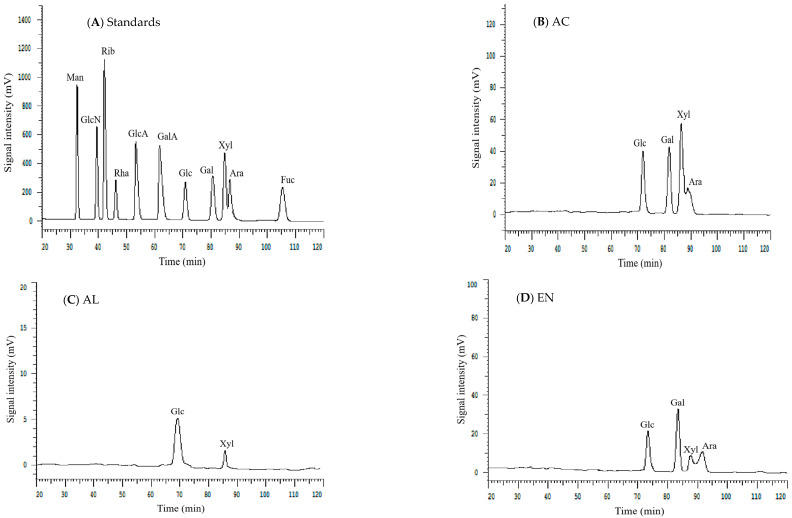
HPLC grams of monosaccharide standards and monosaccharide compositions in Sanchi flower SDF. (**A**) Standards (**B**), AC, Acid (**C**), AL, Alkali (**D**) EN, Enzyme (**E**), H, Homogenization and (**F**), U, Ultrasonication. Mannose, Man; Ribose, Rib; rhamnose, Rha; Glucuronic acid, GlcA; Galacturonic acid, GalA; Glucose, Glc; Galactose, Gal; Xylose, Xyl; Arabinose; Ara; Fucose, Fuc.

**Table 1 foods-11-01995-t001:** SDF yield, protein and color values of Sanchi flower dietary fiber extracted by different methods.

Extraction Methods	SDF Yield	Protein	Color Values		
			L*	a*	b*
AC	11.480 ± 1.440 ^b^	1.400 ± 0.035 ^bc^	65.545 ± 0.177 ^d^	4.110 ± 0.127 ^ab^	19.100 ± 0.127 ^b^
AL	23.140 ± 2.820 ^a^	1.015 ± 0.015 ^c^	66.135 ± 0.078 ^c^	3.945 ± 0.064 ^b^	20.145 ± 0.177 ^a^
EN	9.540 ± 0.240 ^b^	2.765 ± 0.036 ^a^	63.275 ± 0.106 ^e^	4.440 ± 0.170 ^a^	17.460 ± 0.085 ^c^
H	10.530 ± 1.230 ^b^	1.867 ± 0.031 ^b^	67.450 ± 0.198 ^b^	3.750 ± 0.283 ^b^	16.250 ± 0.198 ^d^
U	10.560 ± 0.510 ^b^	2.450 ± 0.025 ^a^	70.120 ± 0.396 ^a^	2.235 ± 0.120 ^c^	18.800 ± 0.339 ^b^

Values are means ± standard deviation of triplicate determinations; ^a–e^ Means followed by different letters in a column are significantly different (*p* < 0.05). L*, lightness; a*, Redness; b*, yellowness. AC, Acid; Al, Alkali; En, Enzyme; H, Homogenization; U, Ultrasonication.

**Table 2 foods-11-01995-t002:** Consistency coefficient, flow behavior index and apparent viscosity of Sanchi flower dietary fiber extracted by different methods.

Extraction Methods	Consistency Coefficient (K)	Flow Behavior Index (n)	Apparent Viscosity 200 1/s (η)	R^2^
U	0.019 ± 0.006 ^c^	3.465 ± 0.096 ^ab^	0.238 ± 0.038 ^b^	0.9998
AL	0.388 ± 0.029 ^b^	3.248 ± 0.485 ^ab^	0.131 ± 0.032 ^c^	0.9995
H	0.008 ± 0.005 ^c^	3.498 ± 0.123 ^a^	0.280 ± 0.014 ^ab^	0.9999
EN	2.404 ± 0.045 ^a^	2.471 ± 0.151 ^b^	0.091 ± 0.036 ^c^	0.9994
AC	0.002 ± 0.001 ^c^	3.724 ± 0.072 ^a^	0.317 ± 0.020 ^a^	0.9995

Values are means ± standard deviation of triplicate determinations; ^a–c^ Means followed by different letters in a column are significantly different (*p* < 0.05). AC, Acid; AL, Alkali; EN, Enzyme; H, Homogenization; U, Ultrasonication.

**Table 3 foods-11-01995-t003:** Molar ratio of monosaccharide components of Sanchi flower soluble dietary fiber extracted by different methods.

Extraction Methods	Glc	Gal	Xyl	Ara
AC	0.77	0.57	1.00	0.52
AL	3.47	-	1.00	-
EN	3.39	3.1	1.00	1.22
H	8.46	5.81	1.00	2.35
U	1.09	0.81	1.00	0.28

Glucose, Glc; Galactose, Gal; Xylose, Xyl; Arabinose; Ara.

**Table 4 foods-11-01995-t004:** Hydration and functional properties of Sanchi flower dietary fiber extracted by different methods.

Extraction Methods	WHC (g/g)	OHC (g/g)	BAC (mg/g)	CAC (mg/g)		GAC (mg/g)
				pH 2	pH 7	
AC	0.820 ± 0.026 ^d^	1.580 ± 0.030 ^c^	3.593 ± 0.010 ^b^	11.607 ± 0.186 ^c^	10.944 ± 0.044 ^d^	73.520 ± 0.416 ^c^
AL	0.890 ± 0.079 ^d^	1.613 ± 0.040 ^c^	3.231 ± 0.004 ^d^	12.725 ± 0.142 ^a^	12.817 ± 0.039 ^a^	83.370 ± 1.022 ^b^
EN	1.167 ± 0.032 ^b^	1.583 ± 0.006 ^c^	3.565 ± 0.006 ^c^	11.864 ± 0.091 ^b^	11.762 ± 0.191 ^c^	51.293 ± 3.916 ^d^
H	1.420 ± 0.061 ^a^	1.827 ± 0.078 ^a^	3.610 ± 0.003 ^a^	12.533 ± 0.065 ^a^	10.425 ± 0.011 ^e^	75.154 ± 0.342 ^c^
U	1.060 ± 0.035 ^c^	1.717 ± 0.038 ^b^	3.618 ± 0.006 ^a^	11.987 ± 0.016 ^b^	12.135 ± 0.080 ^b^	94.626 ± 0.343 ^a^

Values are means ± standard deviation of triplicate determinations; ^a−e^ Means followed by different letters in a column are significantly different (*p* < 0.05). WHC, Water holding capacity; OHC, Oil holding capacity, BAC, Bile acid-adsorption capacity; CAC, Cholesterol-adsorption capacity; GAC, Glucose adsorption capacity.

## Data Availability

Data is contained within the article.

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
