# Peer review of "The Influence of Different Extraction Methods on the Structure, Rheological, Thermal and Functional Properties of Soluble Dietary Fiber from Sanchi (Panax notoginseng) Flower"

_foods, 2022, doi:10.3390/foods11141995_

Round 1
Reviewer 1 Report
In this paper, different extraction methods of soluble dietary fiber from Sanchi flower were introduced. The effect of various extraction methods on microstructural, rheological, thermal, physicochemical and functional characteristics was demonstrated. It is a meaningful work to search for new sources of dietary fiber, but there are doubts in this work.
Abstract
Line 16: the use of abbreviations should be avoided in the abstract. If necessary, abbreviations Glc and Gal need to be explained
Introduction
Lines 41-52: Authors describe dietary fiber as “non-starch polysaccharides that contain a minimum of ten monomeric units, which cannot be hydrolyzed by the enzymes (endogenous) present in the intestines “. However, the objective of this paper is only soluble dietary fiber. I suggest authors to describe the differences between soluble and insoluble fibers and justify why only the soluble dietary fibers were selected for the analysis.
Materials and methods
Please, provide chemical composition of Sanchi flower powders used in study. Without these data it is difficult to understand the values of extraction yields presented in the paper.
Lines 166-178: The description of the measurements of rheological properties is incomprehensible. Chapter 3.5 Rheological properties includes a rheogram (Fig. 2C) showing relationship between shear rate and shear stress and Table 2 showing the rheological characteristics of extracts calculated by the Power Law Model. It therefore remains unclear why the authors submit description of dynamic frequency sweep measurements.
It is stated that „ The viscosity was recorded while increasing the shear rate from 0.10 to 400 s‒1“. However, in the rheogram (Fig. 2C) we can see that the shear rate was up to 1000 1/s.
Please, correct the equation (2) and write the Power Law Model correctly.
Lines 204, 212, 219, 225, 231: Please, indicate the units for the described characteristics.
Results and discussion
Lines 383-396: Author provides such an interpretation of rheological measurements: “However, as shown in Fig 2 (C), the shear rate changes were determined as a function of applied shear stress. The EN-SDF exhibited rapid changes in the shear rate at initial shear force level of 28 Pa. AL-SDF showed modifications in shear rate at applied shear stress level of 55 Pa. On the other hand, U-SDF exhibited more slow gradual changes in shear rate changes when corresponding shear stress was applied at the range of 10-88 Pa. Conversely, H-SDF and AC- SDF exhibited similar rate of change in shear rate as function of corresponding increases in shear stress from 10 to 120 Pa. This could be due to the major structural changes in SDF fibers extracted by H and AL treatments“. However, in describing rheological measurements, the authors indicated the changes in shear rate from 0.10 to 400 s‒1 and measuring the viscosity.
Lines 510-511: “Furthermore, increased GAC values suggest that the SDF has greater capacity during the entire GI transit time.“ Please, elaborate this statement.
Chapter 3. To improve discussion, I suggest the addition of data about the functional properties of soluble dietary fiber extracted from other plants.
Fig. 1: In general, the microscopy analysis presented in the current research is not convincing. Pictures B,D,E very dark and therefore difficult to evaluate.
Figure 2C. It is common for flow curves to be plotted on the x-axis with a shear rate and on the y-axis with a shear stress.
Table 2. You need to show how well the Power law Model fits the experimental data in rheological measurements. Please, provide R2 values. Please, indicate the units for apparent viscosity.
Table 4: Missing units of functional properties.
Author Response
Response to Reviewer 1
In this paper, different extraction methods of soluble dietary fiber from Sanchi flower were introduced. The effect of various extraction methods on microstructural, rheological, thermal, physicochemical and functional characteristics was demonstrated. It is a meaningful work to search for new sources of dietary fiber, but there are doubts in this work.
Abstract
Line 16: the use of abbreviations should be avoided in the abstract. If necessary, abbreviations Glc and Gal need to be explained
As suggested, the abbreviations have been expanded. Line 16.
Introduction
Lines 41-52: Authors describe dietary fiber as “non-starch polysaccharides that contain a minimum of ten monomeric units, which cannot be hydrolyzed by the enzymes (endogenous) present in the intestines “. However, the objective of this paper is only soluble dietary fiber. I suggest authors to describe the differences between soluble and insoluble fibers and justify why only the soluble dietary fibers were selected for the analysis.
As rightly indicated, the differences between SDF and IDF and justification for studying SDF were not be cleared described. Therefore, the differences and justification have been included in Line: 43-49.
Materials and methods
Please, provide chemical composition of Sanchi flower powders used in study. Without these data it is difficult to understand the values of extraction yields presented in the paper.
Apart from protein, constituents of Sanchi flower have been included based on our previous study in Line 267-271.
Lines 166-178: The description of the measurements of rheological properties is incomprehensible. Chapter 3.5 Rheological properties includes a rheogram (Fig. 2C) showing relationship between shear rate and shear stress and Table 2 showing the rheological characteristics of extracts calculated by the Power Law Model. It therefore remains unclear why the authors submit description of dynamic frequency sweep measurements.
For improved comprehension, these lines have been rewritten. The description about dynamic frequency sweep measurements has been removed. Line 180-184.
It is stated that „ The viscosity was recorded while increasing the shear rate from 0.10 to 400 s‒1“. However, in the rheogram (Fig. 2C) we can see that the shear rate was up to 1000 1/s.
This mistake has been corrected. Line 183.
Please, correct the equation (2) and write the Power Law Model correctly.
These mistakes have been corrected. Line 188-190.
Lines 204, 212, 219, 225, 231: Please, indicate the units for the described characteristics.
Units have been included as suggested. Line 216, 223, 230, 238, and 244.
Results and discussion
Lines 383-396: Author provides such an interpretation of rheological measurements: “However, as shown in Fig 2 (C), the shear rate changes were determined as a function of applied shear stress. The EN-SDF exhibited rapid changes in the shear rate at initial shear force level of 28 Pa. AL-SDF showed modifications in shear rate at applied shear stress level of 55 Pa. On the other hand, U-SDF exhibited more slow gradual changes in shear rate changes when corresponding shear stress was applied at the range of 10-88 Pa. Conversely, H-SDF and AC- SDF exhibited similar rate of change in shear rate as function of corresponding increases in shear stress from 10 to 120 Pa. This could be due to the major structural changes in SDF fibers extracted by H and AL treatments“. However, in describing rheological measurements, the authors indicated the changes in shear rate from 0.10 to 400 s‒1 and measuring the viscosity.
This mistake has been corrected. The shear rate change from 0.10 to 400 s‒1 to 0.10 to 1000 s‒1 has been modified. Line 183.
Lines 510-511: “Furthermore, increased GAC values suggest that the SDF has greater capacity during the entire GI transit time.“ Please, elaborate this statement.
This statement has been elaborated. Line 565-567.
Chapter 3. To improve discussion, I suggest the addition of data about the functional properties of soluble dietary fiber extracted from other plants.
Additional data related to SDF from other plants have been included. Line 465-468, 511-513, 525-527, 552-553, 558-560, and 571-572,
Fig. 1: In general, the microscopy analysis presented in the current research is not convincing. Pictures B,D,E very dark and therefore difficult to evaluate.
Regretfully, as we don’t have any samples remaining, it is very difficult to undertake SEM analysis and provide the images. We apologize for not making these suggested changes.
Figure 2C. It is common for flow curves to be plotted on the x-axis with a shear rate and on the y-axis with a shear stress.
As suggested, the values on the X and Y-axis have been swapped and a new graph has been plotted. Line 383.
Table 2. You need to show how well the Power law Model fits the experimental data in rheological measurements. Please, provide R2 values. Please, indicate the units for apparent viscosity.
As suggested, how well experimental data fits the Power law Model has been shown in Table 2. R2 values have been included in a separate column. Line 427.
Table 4: Missing units of functional properties.
Missing units included in Table 4. Line 536.
Thank you for the meticulous review and the invaluable suggestions and guidance.

Reviewer 2 Report
Comments to the Authors
Manuscript ID: foods-1780840
General remarks:
The authors studied the influence of various extraction methods for SDF isolation from novel raw material-Sanchi flower. Performed analysis are relevant in the field of dietary fibres and their further use as food ingredients from technological as well as their health promoting effect. However, several issues are present and must be addressed.
The main drawback is the lack of results regarding dietary fibre content in isolates obtained by different extraction methods. The authors proposed application in food products and in this regard is necessary to know the amount of DF which you have at your disposal for implementation and predicting the technological as well as nutritional aspects of the new product.
Authors in several occasions refer to particle size and specific surface when discussing the results particularly for hydration and functional properties whiteout conducted analysis on the mentioned parameters. According to the extraction methods, grinding or fractioning was not included in the sample preparation process, hence we don’t have any hint on the particle size. Hydration and functional properties indeed depend on the particle size, but relying on this without knowing concrete particle size is not quite relevant. Hence, the authors should also relate these properties with FTIR and ESEM results or perform particle size analysis.
Typographical errors are present in the manuscript, but the quality of the text is rather good and the manuscript is well organized. All tables need revision regarding the letters indicating statistical significance. Figures resolution needs to be improved for clearer visual appearance. and Reference list should be revised to ensure consistency.
The specific comments are listed below.
Abstract
The abstract is well written and reflects all performed analysis and main results.
Manuscript
Lines 43-44: The sentence needs rephrasing for clarity, please improve. Use improves several physiological functions such as…and contributes to disease prevention.
Line 48: Please include that the given values are for adult persons depending on gender.
Line 49: technological food components is not a proper term, please use another such as enhancers, improvers of technological quality.
Line 60: please use drawbacks instead of demerits.
Line 63: Please include that high costs originate from the used enzymes and are increasing with the increasing amount of applied enzymes.
Line 72: Please correct to upper case letter in sanchi flower.
Lines 86: Please include storage conditions.
Line 95: Please include drying conditions.
Line 98: The following equation will give a better insight in the feasibility of extraction process and content of isolated dietary fibre intended for application in food industry.
where m is the mass of SDF obtained from 100 g of Sanchi flower, DFe is the percentage of dietary fibre in SDF extracts, and DFs is the percentage of dietary fibre in Sanchi flower used as raw material.
Line 106: Please include drying conditions and add AL in front of SDF (AL-SDF).
Line 109: Please correct to upper case letter in sanchi flower.
Line 111: The used water was distilled or regular, please indicate. Furthermore, include the chemical used to obtain water of pH 9.5.
Line 112: Please state the used centrifuge for performed centrifugation here, but also in lines 92 and 104.
Line 114: Please include the chemical used for pH adjustment to 7.
Line 117: Please check statement regarding protein solubility at given pH. Isoelectric precipitation of majority of plant proteins is usually performed in the pH range 3.5-4 followed by heating. Hence consider changing the sentence in this regard.
Line 119: Please state the used centrifuge for performed centrifugation.
Line 123: Please use plural. Instead of was place were. Also state is the used water distilled. Check this also throughout the manuscript.
Line 134: Please state the drying conditions.
Line 137: Please state what conversion factors were used for interpretation of N to protein content.
Line 140: Please include that the colour space is CIELab and explain each parameter in + and – value.
Line 146: Please include at which magnification level SEM was performed.
Line 150: Please put in uppercase the adequate parts in the units (cm-1).
Line 149: Please indicate are there any preparations of samples prior to measurement like mixing with KBr and pelleting or the used technique was attenuated total reflectance (ATR).
Line 167: please revise to Peltier plates and include the plate geometry (40 mm, 50 mm or other) and plate surface (rough, smooth or serrated). Also indicate the gap adjustment under which the measurements were performed.
Line 168: Please provide full term for the HIPEs abbreviation.
Line 172: Please put in uppercase the adequate parts in the units, here and check throughout the entire manuscript (line 117).
Line 173: What stands for FG? Please provide the full term abbreviation it is unclear.
Line 176: Please revise the equation. As presented is incorrect. n-1 should be in superscript.
Line 177: -1 and n should be in superscript please revise.
Line 184: Please put the PMP abbreviation here since it was already mentioned before.
Lines 204 and 212: Please check the equations used with other references on the subject. They should express the weight of water/oil retained per gram of fibre (g/g). It seems that here calculations are made based on the excess water/oil not bounded water/oil.
Line 219: Please include the model and manufacturer of the used spectrophotometer.
Line 228: please use singular for concentration, instead of concentrations.
Line 229-230: Please revise typographical error for °C, and exclude x from the centrifugation conditions since in sections above it was just 5000 g. Keep that consistent throughout the manuscript.
Line 231: Please state the exact glucose kit used in the analysis.
Line 233: Triplicate should be with small t. Please revise.
Line 256: Here p value was given with small letter also in line 267 while in line 244 is given as P. Please decide and uniform in entire manuscript.
Line 258: Please relate the parameters a* and b* with corresponding colours here in discussion and make easier to readers to understand the outcomes of this analysis.
Line 264: Please connect the samples in the discussion with appropriate letter on the figure, such as sample AC-SDF (figure 1A) etc. All figures are very dark and a bit blurry so try to enhance that by adjusting brightness and sharpness.
Line 276: please use superscript for wavenumber unit in the entire section (lines 279, 281, 284, 334, 336, 341, 344, 348).
Line 354: Celsius degree was separated from the number in the previous part of the manuscript, here is separated. Please uniform in entire manuscript.
Line 387: Please revise the sentence for better understanding.
Line 398: Please include references for the mentioned studies.
Line 410: Extraction method is enough, please exclude type.
Line 442: Please include soluble dietary fibre in the Table title.
Line 444: Since it is supposed that cellulose is degraded shouldn’t that increase the glucose content? Please check by consulting more literature. What about starch?
Line 467: Connect WHC and OHC with ESEM and FTIR results.
Line 472: Please write it can be utilized…
Line 476: Please add reference of the corresponding studies.
Line 488: Functional properties depend on the particle size, but relying on this without knowing concrete particle size is not quite relevant. Hence, the authors should also relate these properties with FTIR results or perform particle size analysis.
Line 502: Please add appropriate reference here.
Line 503: Did the authors meant could be related to?
Line 521: Please state the full names of the extraction methods.
Line 522: For is typed twice. Please revise.
Line 537: Sanchi is written with small s. Uniform throughout the manuscript.
Tables – All tables need to be revised in terms of putting letters denoting statistical significance in superscript.
Figures – Pictures in better quality are to be provided by the authors in all figures.
Author Response
REVIEWER 2
General remarks:
The authors studied the influence of various extraction methods for SDF isolation from novel raw material-Sanchi flower. Performed analysis are relevant in the field of dietary fibres and their further use as food ingredients from technological as well as their health promoting effect. However, several issues are present and must be addressed.
The main drawback is the lack of results regarding dietary fibre content in isolates obtained by different extraction methods. The authors proposed application in food products and in this regard is necessary to know the amount of DF which you have at your disposal for implementation and predicting the technological as well as nutritional aspects of the new product.
Composition of Sanchi flower DF based on our previous studies has been included.
Authors in several occasions refer to particle size and specific surface when discussing the results particularly for hydration and functional properties whiteout conducted analysis on the mentioned parameters. According to the extraction methods, grinding or fractioning was not included in the sample preparation process, hence we don’t have any hint on the particle size. Hydration and functional properties indeed depend on the particle size, but relying on this without knowing concrete particle size is not quite relevant. Hence, the authors should also relate these properties with FTIR and ESEM results or perform particle size analysis.
It was inferred from other studies wherein a particular extraction method caused changes in the particle size. Surface area and particle size are included with other factors, which influence physicochemical and functional properties. However, as suggested, discussions about the link between functional properties and FTIR and SEM results have been included (particularly with WHC & OHC). Unfortunately, due to lack of availability of Sanchi SDF samples, particle size analysis cannot be undertaken. Nonetheless, brief descriptions have been included.
Typographical errors are present in the manuscript, but the quality of the text is rather good and the manuscript is well organized. All tables need revision regarding the letters indicating statistical significance. Figures resolution needs to be improved for clearer visual appearance. and Reference list should be revised to ensure consistency.
Typos have been corrected. Text and figures have been corrected. Reference list has been revised.
The specific comments are listed below.
Abstract
The abstract is well written and reflects all performed analysis and main results.
Manuscript
Lines 43-44: The sentence needs rephrasing for clarity, please improve. Use improves several physiological functions such as…and contributes to disease prevention.
This sentence has been rewritten for improved clarity. Line 47-49.
Line 48: Please include that the given values are for adult persons depending on gender.
Given values are for an adult. This sentence has been modified. Line 53.
Line 49: technological food components is not a proper term, please use another such as enhancers, improvers of technological quality.
This sentence has been modified as suggested. Line 54-55.
Line 60: please use drawbacks instead of demerits.
Demerits replaced with drawbacks. Line 65.
Line 63: Please include that high costs originate from the used enzymes and are increasing with the increasing amount of applied enzymes.
Increasing costs associated with increasing enzyme amount has been included. Line 68-69.
Line 72: Please correct to upper case letter in sanchi flower.
Sanchi has been corrected to upper case. Line 78.
Lines 86: Please include storage conditions.
Storage conditions included. Line 92.
Line 95: Please include drying conditions.
Drying conditions included. Line 101-102.
Line 98: The following equation will give a better insight in the feasibility of extraction process and content of isolated dietary fibre intended for application in food industry.
where m is the mass of SDF obtained from 100 g of Sanchi flower, DFe is the percentage of dietary fibre in SDF extracts, and DFs is the percentage of dietary fibre in Sanchi flower used as raw material.
Regretfully, as we could not find the aforementioned formula in any references, it was not included. We apologize for not having made this particular change in the manuscript.
Line 106: Please include drying conditions and add AL in front of SDF (AL-SDF).
Drying conditions included. AL added before SDF. Line 112.
Line 109: Please correct to upper case letter in sanchi flower.
Sanchi has been corrected to upper case. Line 115.
Line 111: The used water was distilled or regular, please indicate. Furthermore, include the chemical used to obtain water of pH 9.5.
Chemical used for pH adjustment has been included. Line 117.
Line 112: Please state the used centrifuge for performed centrifugation here, but also in lines 92 and 104.
These lines have been modified as suggested. Line 98-99, 110, and 118-119.
Line 114: Please include the chemical used for pH adjustment to 7.
Chemical used for pH adjustment has been included. Line 121.
Line 117: Please check statement regarding protein solubility at given pH. Isoelectric precipitation of majority of plant proteins is usually performed in the pH range 3.5-4 followed by heating. Hence consider changing the sentence in this regard.
These lines have been modified as suggested. Line 124-125.
Line 119: Please state the used centrifuge for performed centrifugation.
These lines have been modified as suggested. Line 126.
Line 123: Please use plural. Instead of was place were. Also state is the used water distilled. Check this also throughout the manuscript.
As rightly pointed, there should have been a plural word. This mistake has been corrected. Line 131.
Line 134: Please state the drying conditions.
Drying conditions included. Line 142.
Line 137: Please state what conversion factors were used for interpretation of N to protein content.
Conversion factor has been included. Line 146.
Line 140: Please include that the colour space is CIELab and explain each parameter in + and – value.
CIElab color space information has been included. Line 149-152.
Line 146: Please include at which magnification level SEM was performed.
Magnification levels included. Line 158.
Line 150: Please put in uppercase the adequate parts in the units (cm-1).
These mistake have been corrected. Line 162.
Line 149: Please indicate are there any preparations of samples prior to measurement like mixing with KBr and pelleting or the used technique was attenuated total reflectance (ATR).
Spectra were obtained in Attenuated Total Reflection (ATR) mode in the wavelength range of 4000–400 cm−1. This has been included. Line 163.
Line 167: please revise to Peltier plates and include the plate geometry (40 mm, 50 mm or other) and plate surface (rough, smooth or serrated). Also indicate the gap adjustment under which the measurements were performed.
There were few mistakes in this section. Therefore, it has been rewritten. Line 180-181.
Line 168: Please provide full term for the HIPEs abbreviation.
As explained above, these lines have been rewritten. Line 182.
Line 172: Please put in uppercase the adequate parts in the units, here and check throughout the entire manuscript (line 117).
These mistakes have been corrected. Line 182-183
Line 173: What stands for FG? Please provide the full term abbreviation it is unclear.
This mistake has been corrected. FG removed.
Line 176: Please revise the equation. As presented is incorrect. n-1 should be in superscript.
This mistake has been corrected. Line 188.
Line 177: -1 and n should be in superscript please revise.
This mistake has been corrected. Line 189.
Line 184: Please put the PMP abbreviation here since it was already mentioned before.
Only abbreviation is included. Line 196.
Lines 204 and 212: Please check the equations used with other references on the subject. They should express the weight of water/oil retained per gram of fibre (g/g). It seems that here calculations are made based on the excess water/oil not bounded water/oil.
OHC and WHC formulae have been changed to express the weight of oil/water retained per gram of fibre (g/g). Line 216 and 223.
Line 219: Please include the model and manufacturer of the used spectrophotometer.
Model and manufacturer of the spectrophotometer included. Line 229-230.
Line 228: please use singular for concentration, instead of concentrations.
It has been changed as suggested. Line 240.
Line 229-230: Please revise typographical error for °C, and exclude x from the centrifugation conditions since in sections above it was just 5000 g. Keep that consistent throughout the manuscript.
This mistake has been corrected. Line 126, 221, and 242.
Line 231: Please state the exact glucose kit used in the analysis.
Details about glucose kit included. Line 243.
Line 233: Triplicate should be with small t. Please revise.
This mistake has been corrected. Line 246.
Line 256: Here p value was given with small letter also in line 267 while in line 244 is given as P. Please decide and uniform in entire manuscript.
These inconsistencies have been removed. Line 257, 261, 353, 429, and 539.
Line 258: Please relate the parameters a* and b* with corresponding colours here in discussion and make easier to readers to understand the outcomes of this analysis.
The corresponding colors (text) have been included. Line 276.
Line 264: Please connect the samples in the discussion with appropriate letter on the figure, such as sample AC-SDF (figure 1A) etc. All figures are very dark and a bit blurry so try to enhance that by adjusting brightness and sharpness.
Discussions have been appropriately linked to the sample image labels. Line 282-283.
Line 276: please use superscript for wavenumber unit in the entire section (lines 279, 281, 284, 334, 336, 341, 344, 348).
Wavenumber unit has been modified to superscript in this entire section and throughout the manuscript. Line 321, 324, 326, 333, 339, 341, and 345.
Line 354: Celsius degree was separated from the number in the previous part of the manuscript, here is separated. Please uniform in entire manuscript.
This inconsistencies have been removed.
Line 387: Please revise the sentence for better understanding.
For improved understanding, this sentence is revised. Line 411-413.
Line 398: Please include references for the mentioned studies.
Reference has been included. Line 417.
Line 410: Extraction method is enough, please exclude type.
As rightly pointed, method is not needed. Line 432.
Line 442: Please include soluble dietary fibre in the Table title.
“soluble” has been included in the table title. Line 505.
Line 444: Since it is supposed that cellulose is degraded shouldn’t that increase the glucose content? Please check by consulting more literature. What about starch?
As rightly pointed, increase in glucose content is related to higher hydrolysis of cell wall cellulose. Therefore, lower glucose could be due to a decrement in hydrolysis by that particular extraction method. Starch can also be a source of glucose. Hence, the sentences have been modified. Line 435-442.
Line 467: Connect WHC and OHC with ESEM and FTIR results.
WHC and OHC have been linked with ESEM and FTIR results along with other factors. Line 465 -511.
Line 472: Please write it can be utilized…
It has been changed as suggested. Line 515.
Line 476: Please add reference of the corresponding studies.
Reference has been added. Line 520.
Line 488: Functional properties depend on the particle size, but relying on this without knowing concrete particle size is not quite relevant. Hence, the authors should also relate these properties with FTIR results or perform particle size analysis.
It was inferred that changes in absorption peaks could have affected the particle size and surface characteristics of SDF. However, as per your suggestion, a brief description about FTIR spectra on particle surface characteristics has been included. Moreover, as no particle size analysis is provided, this sentence has been removed and lines have been included. Line 525-530.
Line 502: Please add appropriate reference here.
Appropriate reference has been added. Line 555.
Line 503: Did the authors meant could be related to?
This sentence has been modified. Line 556.
Line 521: Please state the full names of the extraction methods.
Full names were added. Line 579-580.
Line 522: For is typed twice. Please revise.
The mistake has been corrected. Line 580.
Line 537: Sanchi is written with small s. Uniform throughout the manuscript.
The mistake has been corrected. Line 595.
Tables – All tables need to be revised in terms of putting letters denoting statistical significance in superscript.
Letters in all the tables have been included as superscripts.
Thank you for the meticulous review and the invaluable suggestions and guidance.

Reviewer 3 Report
Legend text is unclear in the graphics Fig. 2, it is important standardize the letter type and size.
Fig. 3, the text of the X and Y axis is not clear lack of sharpness. What does the letter A mean?, is not specified in the corresponding graph.
It is recommended to specify the meaning of the acronyms CAC, GAC, as well as the rest of the methods used below the tables, although at the beginning is indicated, sometimes its meaning can be lost as the reading of the document progresses article.
Author Response
Response to Reviewer 3
Legend text is unclear in the graphics Fig. 2, it is important standardize the letter type and size.
Clarity of Fig.2 has been improved and letter type and size standardized. Line 372.
Fig. 3, the text of the X and Y axis is not clear lack of sharpness. What does the letter A mean?, is not specified in the corresponding graph.
Letter A has been removed from Fig 3. The clarity of this figure has been improved. Line 475.
It is recommended to specify the meaning of the acronyms CAC, GAC, as well as the rest of the methods used below the tables, although at the beginning is indicated, sometimes its meaning can be lost as the reading of the document progresses article.
Abbreviations have expanded and included in all the table and figures. Line 257, 310, 392, 429, 504, 507, and 540.
Thank you for the meticulous review and the invaluable suggestions and guidance.

Round 2
Reviewer 2 Report
Respected authors,
Thank You for accepting the proposed suggestions and performing manuscript revision in adequate manner.
I appreciate their cooperation regarding manuscript improvements.
Only minor changes are needed and final revision for checking hidden typographical errors. Listed below.
Line 36: please write H2O2.
Line 225: please change to “the excess oil has been removed”. All oil is incorrect.
Line 238: full stop is missing at the end of the sentence. Please add. Line 244 also.
Line 261: Please check. I think it has to be p < 0.05.
Line 270: Please state all mineral elements with small first letter.
Author Response
Response to Reviewer 2
Thank You for accepting the proposed suggestions and performing manuscript revision in adequate manner.
I appreciate their cooperation regarding manuscript improvements.
Only minor changes are needed and final revision for checking hidden typographical errors. Listed below.
Line 36: please write H2O2.
Thank you for pointing out this mistake. Subscripts have been included.
Line 225: please change to “the excess oil has been removed”. All oil is incorrect.
Thank you for pointing out this mistake. As suggested, “All oil” has been changed to “excess oil”.
Line 238: full stop is missing at the end of the sentence. Please add. Line 244 also.
Full stop has been added.
Line 261: Please check. I think it has to be p < 0.05.
We meant to state that the different extraction method had similar (no difference) yields or statistically insignificant yields (p > 0.05). Line 261-262.
Line 270: Please state all mineral elements with small first letter.
First letters of mineral elements have been changed to lower case.
Thank you for the meticulous correction. We are very grateful for your suggestions and guidance. We have learnt a great deal from your step-by-step careful review.
